# Biodentine Inhibits the Initial Microbial Adhesion of Oral Microbiota In Vivo

**DOI:** 10.3390/antibiotics12010004

**Published:** 2022-12-20

**Authors:** Ali Al-Ahmad, Michael Haendel, Markus Joerg Altenburger, Lamprini Karygianni, Elmar Hellwig, Karl Thomas Wrbas, Kirstin Vach, Christian Tennert

**Affiliations:** 1Medical Center, Department of Operative Dentistry and Periodontology, Faculty of Medicine, University of Freiburg, Hugstetter Str. 55, 79106 Freiburg, Germany; 2Clinic of Conservative and Preventive Dentistry, University of Zurich, Plattenstrasse 11, 8032 Zurich, Switzerland; 3Institute of Medical Biometry and Statistics, Faculty of Medicine, University of Freiburg, Stefan-Meier-Str. 26, 79104 Freiburg, Germany; 4Department of Restorative, Preventive and Pediatric Dentistry, University of Bern, Freiburgstrasse 7, 3010 Bern, Switzerland

**Keywords:** bioactive material, bioactive ceramics, Biodentine, MTA, AH Plus, microoganisms, initial adhesion, endodontics, calcium silicate

## Abstract

This study aimed to evaluate the in vivo initial microbial adhesion of oral microorganisms on the biomaterial Biodentine compared to MTA and AH Plus. Cylindrical samples of the materials were prepared, and dentin slabs served as a control. An individual intraoral lower jaw splint served as a carrier for the samples and was worn by six volunteers. The specimens were worn for 120 min. Adherent bacteria were quantified by determining the colony-forming units (CFUs), while the visualization and quantification of total adherent microorganisms were facilitated by using DAPI and live/dead staining combined with fluorescence microscopy. Bovine dentin had a significantly higher number of aerobic CFUs compared to Biodentine (*p* = 0.017) and MTA (*p* = 0.013). The lowest amounts of DAPI-stained adherent microorganisms were quantified for Biodentine (15% ± 9%) and the control (18% ± 9%), while MTA showed the highest counts of initially adherent microorganisms (38% ± 10%). Significant differences were found for MTA and Biodentine (*p* = 0.004) as well as for MTA and the control (*p* = 0.021) and for AH Plus and the control (*p* = 0.025). Biodentine inhibited microbial adherence, thereby yielding an antimicrobial effectivity similar to that of MTA.

## 1. Introduction

In endodontics, dental calcium silicate types of cement have been successfully used for many years for root repair and treatments aiming at the maintenance of pulp vitality. Silicate cements, also known as bioactive bioceramics, or hydraulic silicates are superior to other endodontic cements which include better biocompatibility, bioactivity, improved sealing, and hydrophilicity [1,2]. During the setting reaction of calcium silicate cement, calcium hydroxide is released and contributes to its high biocompatibility and bioactivity [3,4]. Calcium silicate cements are resorption-resistant, antimicrobial, and bone-inductive sealing materials, which mainly promote the formation of new hard tissue [3,5,6]. Several studies have found high success rates of MTA use as a sealing or root filling material in endodontic treatment [7,8,9]. Biodentine, a dentin replacement material that belongs to the group of calcium silicate cements, has drawn attention over the last decade. It has a wide range of applications and can be used in the coronal tooth region as a relining cement or temporary filling material, and also as a endodontic material for indirect and direct pulp capping [7,10]. The improved mechanical properties of Biodentine, such as its compressive and flexural strength, as well as its reduced setting time have been significantly improved compared to those of MTA. Other material properties of Biodentine, such as its biocompatibility, bioactivity, color stability, microleakage, solubility, and dentin adhesion have also been investigated to date [11,12,13,14]. MTA and Biodentine have been found to have similar cytotoxic effects [15,16,17,18].

Caries treatment aims to completely remove all carious tissues and adherent microorganisms to prevent further cariogenic activity and provide well-mineralized dentin for restoration [19]. The success of restorative treatment, focusing on the complete elimination of bacteria and complete caries removal, is still considered the most common treatment strategy, irrespective of the restorative material used [20]. However, modern treatment protocols allow for partial caries removal in deep lesions to reduce the risk of pulp exposure [21,22]. In this case, partial removal of carious tissue aims at maintaining the deeper layer of infected carious dentin, which has the ability to remineralize [21]. Partial caries removal is favorable in terms of carious lesion progression and longevity of the restorations, as well as the preservation of pulpal tissue [22]. Even after total caries removal and the use of antiseptics in deep lesions, microorganisms were found to be present in residual dentin and dentinal tubules. Biodentine is placed in lesions in which microorganisms are present and thus its antimicrobial potential is of great importance [21,23,24]. Interestingly, MTA was shown to have low antimicrobial activity against *Enterococcus faecalis in a previous report* [25]. In previous in vitro studies, Biodentine showed a higher antimicrobial activity compared to MTA and intermediate restorative material [23,24]. However, to date there has been no study evaluating the antimicrobial properties of Biodentine in vivo. The present in-vivo study aimed to evaluate the initial adhesion of oral microorganisms on Biodentine compared to MTA, AH Plus, and bovine dentin (control).

## 2. Results

### 2.1. Microbial Adhesion on Biodentine, MTA, AH Plus, and Bovine Dentin

Figure 1A,B show the microbial growth rates of initially adherent oral aerobic (Figure 1A) and anaerobic (Figure 1B) microorganisms after a two-hour exposure in the oral cavity. Bovine dentin served as a negative control, while AH Plus, Biodentine, and MTA served as the material test surfaces. Additionally, Table 1 presents the distribution of the aerobic and anaerobic microorganisms in relation to the slab surface area. Each material sample yielded a percentage of adherent microorganisms. For the six study participants an average was calculated for each material before the standard deviation was calculated. Statistically significant differences were only found for aerobic microorganisms, while the differences between the anaerobic microorganisms were not significant (*p* > 0.5). In particular, bovine dentin (mean ± standard deviation; 53.2% ± 32.0%) allowed for the adhesion of a significantly larger amount of aerobic microorganisms compared to Biodentine (10.5% ± 15.7%, *p* = 0.017) and MTA (8.5% ± 12.4%, *p* = 0.013) (Table 1, Figure 1A,B). 

### 2.2. Fluorescence Microscopic Evaluation with DAPI

Table 2 presents an overview of the material surface coverage (in%) with initially adherent microorganisms after DAPI staining and fluorescence microscopy, while Figure 2 shows the fluorescence microscopic images of the DAPI-stained sample surfaces of AH Plus, Biodentine, MTA, and bovine dentin (control).

The lowest amounts of DAPI-stained adherent microorganisms were quantified for Biodentine (mean ± standard deviation; 15% ± 9%) and the control (18% ± 9%), while MTA showed the highest counts of initially adherent microorganisms (38% ± 10%). Significant differences were found for MTA and Biodentine (*p* = 0.004) as well as for MTA and the control (*p* = 0.021) and for AH Plus and the control (*p* = 0.025), respectively (Table 2, Figure 2). The representative fluorescence microscopy images after DAPI staining are shown in Figure 3. 

### 2.3. Live/Dead Staining

Table 3 gives an overview of the material surface coverage (in%) with initially adherent microorganisms after live/dead staining and fluorescence microscopy, while Figure 4 demonstrates the fluorescence microscopic images of the live/dead-stained sample surfaces of AH Plus, Biodentine, MTA, and bovine dentin (control).

The proportion of living (active) and dead (non-active) microorganisms was evaluated in relation to the surface area of the material and control samples and the calculations yielded a percentage value for each subject and material sample. All material and dentin slabs revealed live and dead microorganisms on their surfaces. Statistically significant higher proportions of live microorganisms were found on the surfaces of AH Plus (34% ± 7%) and the control (37% ± 18%) compared to the bioactive materials Biodentine (15% ± 9%) and MTA (14% ± 8%) (Table 3, Figure 4A). Looking at the proportions of non-active bacteria, bovine dentin (4%) and AH Plus (12%) showed the lowest amount of non-active bacteria. Slightly higher proportions of non-active bacteria were found on Biodentine (16%), while the highest proportions were found for MTA (68%, *p* < 0.001) (Table 3, Figure 4B). Representative fluorescence microscopy images after live/dead staining are presented in Figure 5.

## 3. Discussion

In the present study, significant differences in the initial adhesion of oral microbiota (2 h) were found for the materials tested (AH Plus, Biodentine, MTA). The control, namely bovine dentin, seems to be the most favorable surface for microbial adhesion and microbial viability among the materials tested. In contrast, MTA and Biodentine appear to have an antimicrobial effect, since both materials showed a significant reduction in bacterial viability as was revealed by the determination of the CFUs. Interestingly, MTA showed the largest amount of non-active microorganisms on its surface. However, significantly more oral microorganisms adhered to MTA compared to Biodentine. Previous in vitro studies investigated the antimicrobial effect of calcium silicate cements and other endodontic materials against several microbial strains. Several studies found the strongest antibacterial activity of Biodentine against *Streptococcus sanguis* strains, significantly higher than MTA and intermediate restorative material in vitro [23,24,26]. In contrast, minimal or almost no antibacterial activity of Biodentine was seen against *Streptococcus mutans* for Biodentine, while MTA had significantly higher antimicrobial activity in vitro [23,24,26]. *Streptococcus sanguis* and *Streptococcus mutans* are common oral microorganism, that have cariogenic properties [27]. A higher antimicrobial activity aginst *Enterococcus faecalis* and *Escherichia coli* was found for Biodentine compared to MTA, in particular ProRoot MTA and MTA Plus [5,6,28]. 

The significant difference in initial adhesion and distribution may be caused by the material-specific characteristics and/or roughness of the surfaces. After extra-oral polishing of the material surfaces with a mean of 0.2 μm, an identical initial roughness of the polished materials could be achieved [29]. The subsequent intraoral exposure of the samples for 120 min may cause a material-specific interaction with the environment, which probably results in changes to the material surface roughness. A previous study investigated biomaterials in liquid solutions and showed that when Biodentine was placed in a phosphate-buffered saline solution (PBS) for one hour or 24 h, a weight loss of the sample was observed, whereas a weight gain was found for MTA [13]. This indicates an increased absorbency of MTA, which may result in a change in the surface roughness. A previous study investigated the influence of different media, such as PBS, blood, an acidic environment, and dry storage on the surface roughness of Biodentine and MTA [30]. The authors also found a slightly increased surface roughness for MTA in PBS after the first 45 min compared to Biodentine and assumed that proteins and minerals contained in the material MTA could bind to its surface and change its profile when it is stored in different liquids [31]. Concerning the porosity of silicate cements, both Biodentine and MTA (ProRoot MTA) demonstrated low average pore diameter, porosity and total pore area compared to [32,33]. Two studies found an even lower porosity of Biodentine compared to MTA which directly influences the adhesion behavior of different microorganisms, [33,34] which led to the hypothesis that the reduced porosity of Biodentine is related to the lower liquid/powder ratio. Especially in the first hours of biofilm formation, the physiochemical material properties influence both pellicle formation and bacterial adhesion [35]. It should also be mentioned that in oral niches such as deep tooth cavities or endodontic applications of various types of cement, a constant salivary flow is normally not to be expected, which leads to a reduction in the shear forces and possibly to an increased influence of intermolecular forces during biofilm formation.

Interestingly, the bioactive materials MTA and Biodentine showed antimicrobial properties in the present study. Live/dead staining combined with fluorescence microscopy revealed a significantly higher number of live microorganisms on AH Plus and bovine dentin compared to MTA and Biodentine whereby a significantly higher proportion of dead bacteria was detected on MTA compared to all other material surfaces. As already described in the literature, the hydroxyl ions of MTA released during the setting process by dissociation of calcium hydroxide have a strong basic effect and presumably lead to the destruction of the bacterial membrane integrity, denaturation of its proteins, and DNA damage [36]. For Biodentine and MTA, several studies have demonstrated a high release of calcium and hydroxyl ions, which inevitably lead to a strong alkalization of the environment [12,33]. In a previous study, a time-dependent alkalizing activity of MTA and Biodentine was investigated in distilled water in vitro [33]. The authors found that Biodentine and MTA had an alkaline pH of 11.6/10.99 after 3 h, which decreased to a pH of 9.26/7.20 after 28 days. This suggests that Biodentine has a stronger and longer antibacterial effect than MTA. However, this effect could not be confirmed in the present study, because in addition to its limited duration (2 h), the experiment was carried out in the oral cavity under constant salivary flow, which probably prevented the development of an alkaline environment around the material samples. According to previous findings, the aluminum oxide found only in MTA could explain the increased number of dead bacterial cells on MTA [12]. Nevertheless, it should be emphasized that significantly fewer microorganisms adhered to Biodentine than to MTA. This should be considered an advantage since “new colonizers” were able to adhere to the dead bacterial cells, which were shown to be adherent to MTA.

In the literature, only a limited number of studies have been carried out on the antimicrobial effect of MTA and Biodentine. Previous in vitro studies investigated the antimicrobial effect of calcium silicate cements and other endodontic materials against several microbial strains. Several studies found the strongest antibacterial activity of Biodentine against *Streptococcus sanguis* strains, significantly higher than MTA and intermediate restorative material in vitro [23,24,26]. In contrast, minimal or almost no antibacterial activity of Biodentine was seen against *Streptococcus mutans* for Biodentine, while MTA had significantly higher antimicrobial activity in vitro [23,24,26]. *Streptococcus sanguis* and *Streptococcus mutans* are common oral microorganism, that have cariogenic properties [27]. A higher antimicrobial activity aginst *Enterococcus faecalis* and *Escherichia coli* was found for Biodentine compared to MTA, in particular ProRoot MTA and MTA Plus [5,6,28]. One in vivo study found a beneficial antimicrobial effect when Biodentine was applied to residual carious dentin compared [37]. These findings in line with the results of the present study.

A limitation using silicate cements is the long setting time of up to 3 h for MTA and approximately 45 min for Biodentine [10,38]. The presence of water is necessary for setting and there is a potential of MTA being washed out whenever there is a communication between the oral cavity and the perforation [39]. Using calcium silicates in endodontics and as a dentine replacement, discolorations of the regarding tooth may occur [40]. The mechanism of tooth discoloration was found to be a result of oxidation of the heavy metal oxides (i.e., iron, bismuth or manganese) contained in calcium silicate cements [41]. Although the color change was greater with gray MTA, both, gray and the modified formula of white MTA without iron and manganese induced clinically perceptible crown discoloration [42]. Biodentine contains Zirconium oxide as radiopacifier and doesn’t cause tooth discoloration. However, in contact with sodiumhypochlorite, chlorhexidine or blood discolorations may occur [40]. Additionally, retreatment of endodontic treatments in cases when calcium silicate cements were used as root canal filling material is very difficult [3]. Further research is necessary to evaluate the appropriate concentration, application method, material layer thickness and possibly necessary pretreatments of the application area for the bioactive materials. There is no report on the long-term clinical performance of these tricalcium silicate cement sealers in the literature.

In the present in vivo study, intraoral splints served as a carrier for the material, and the control surfaces and similar splints have been used in previous own studies [43,44,45]. In this way, oral biofilm formation can be investigated and the structural integrity of the intraorally formed in situ biofilm can be maintained until its extraoral analysis. The arrangement of the individual material samples buccally as well as the positioning of the splint in the maxilla or mandible cannot significantly influence the local biofilm formation [43]. Obviously, in situ studies of microbial adhesion and biofilm formation on intraoral hard structures are clearly preferable to in vitro studies [35,46]. The main reasons for this are the complex intraoral conditions, such as microbial diversity and interaction, and masticatory and abrasive forces emanating from the tongue muscles, cheeks, lips, and rinsing function of saliva, but also antimicrobial substances in food, saliva, and dental care products.

In the present study, a total of six participants carried the intraoral splints as performed previously [47]. However, future studies should aim for a lager sample size since would improve the validity of the findings and potential cofounding factors.

The present work aimed to quantify and compare the microbial adhesion of the initial biofilm (2 h) on different materials (Biodentine, MTA, AH Plus). Compared to immunological, molecular, or microscopic methods, the culture method has the advantage of direct bacterial vitality detection and the determination of its proliferation ability. However, dead cells and other abiotic structures cannot be quantified [48]. The culture method is also not limited by a higher number of bacteria, as the number of colonies per culture medium can be adjusted by means of dilution series for a clear quantification. The cultivation of bacterial colonies on selective culture media has been performed in many previous studies [48,49,50].

## 4. Materials and Methods

### 4.1. Subject Recruitment

Prior to the recruitment of the subjects, the study was approved by the University of Freiburg Ethics committee (Reference number 311/14). Participants were recruited from the dental students of the University of Freiburg—Medical Center, Germany. 

#### 4.1.1. Inclusion Criteria


-age ≥ 18 years-caries-free


#### 4.1.2. Exclusion Criteria


-known allergy to the materials or their components and/or suffering from infectious or life-threatening diseases-pregnancy or breastfeeding-temporary use of antibiotics in the last six months or anti-inflammatory medication within the last 30 days-serious general illnesses, such as diabetes, HIV, hepatitis B and C, acute tumor diseases, or epilepsy


A total of six participants were recruited for the study. Prior to the experiments, a clinical oral examination was performed. The subjects showed no signs of gingivitis or caries. Informed written consent was provided by the volunteers for participation in the study and the participants could terminate their participation in the study at any time without giving any reasons.

### 4.2. Material Samples

Cylindrical enamel-dentin slabs (diameter 5 mm, 19.63 mm^2^ surface area, height 1.5 mm) were prepared from the labial surfaces of bovine incisors from 2-year-old BSE-negative cattle. The BSE status was tested by the veterinary unit at the slaughterhouse using the IDEXX laboratories BSE diagnostic kit (Ludwigsburg, Germany). For the fabrication of the dentin slabs, the enamel was removed and the outer dentin surface was used for the oral exposure experiments. The surfaces of all samples were polished by wet grinding with abrasive paper (400 to 4000 grit) as described in a previous study [44].

The bovine dentin slabs were disinfected by ultrasonication in 70% ethanol for 1 min, followed by air-drying and 1 min ultrasonication in 20% EDTA (Pharmacy of the University of Freiburg, Medical Center, Germany) to remove the smear layer. Afterward, the slabs were washed twice for 5 min in double distilled water. The enamel samples were then treated by ultrasonication for 1 min in 3% NaOCl (Aug. Hedlinger GmbH, Stuttgart, Germany) to remove the superficial smear layer [44,51]. Subsequently, 10 min ultrasonication in 70% ethanol (Sigma-Aldrich Chemie GmbH, Steinheim, Germany) was carried out for disinfection, followed by 10 min ultrasonication in double distilled water. The bovine dentin slabs were stored in distilled water for 24 h for hydration before exposure in the oral cavity [52]. Subsequently, the samples were free of bacteria as confirmed by the determination of the CFUs after desorption.

The dental material samples Biodentine (Biodentine™, Septodont, Saint-Maur-des-Fossés Cedex, France), AH Plus (AH Plus^®^, Dentsply Sirona, Bensheim, Germany), and MTA (ProRoot^®^ MTA, Dentsply Sirona, Bensheim, Germany) were prepared according to the manufacturer’s instructions using sterile instruments. After mixing the components of each material, a mold made of rigid silicone (Aquasil Hard Putty, Dentsply Sirona, Bensheim, Germany) was used to achieve a final sample size of 5 mm in diameter, 19.63 mm^2^ surface area, and 1.5 mm in height. After setting, the surfaces of all samples were polished by wet grinding with abrasive paper (400 to 4000 grit). The dental material slabs were finally disinfected by immersion in 70% ethanol for 3 sec followed by washing for 10 min in double distilled water.

### 4.3. Intraoral Splints

An individual intraoral lower jaw splint that served as a carrier for the material samples was prepared for each volunteer (Figure 6). This procedure for the in situ examination of the oral biofilm has been described previously [44,47,51]. Before intraoral use, each splint was disinfected by 1 min ultrasonication in 70% ethanol. The slabs were fixed onto individual lower jaw splints with a polysiloxane impression material (Aquasil Ultra, Dentsply DeTrey GmbH, Konstanz, Germany), as described previously [44]. This ensured that only the surface of the slabs was exposed to the oral cavity, as the margins were completely covered by the impression material. The specimens were fixed at the buccal sites of the lower premolars and the first molar and carried for 120 min, respectively (Figure 6). In total, each subject carried the splint three times, since the adhered bacteria were determined using three different microbiological methods: colony forming units (CFU), live/dead staining, and DAPI (4′,6-Diamidine-2′-phenylindole dihydrochloride) staining combined with fluorescence microscopy. In each run, the splint contained one bovine enamel slab, one AH Plus sample, two Biodentine samples, and two MTA samples. 

### 4.4. Determination of the Colony Forming Units (CFUs)

The CFUs were quantified as described in detail earlier [53]. In brief, after exposure in the oral cavity, the specimens were rinsed for 10 sec each with 1 mL sterile 0.9% NaCl to remove non-adherent microorganisms. Additionally, the reverse material surfaces and their upright side margins were cleansed using small, sterile foam pellets (Voco GmbH, Cuxhaven, Germany) after which each material was transferred into sterile Eppendorf tubes (Eppendorf GmbH, Wesseling-Berzdorf, Germany) and ultrasonicated for 2 min in 1 mL 0.9% NaCl on ice. After vortexing for 30–45 s, the suspensions of each material were then serially diluted up to 1:10^3^ in 0.9% NaCl. Aerobic and facultative anaerobic bacteria were cultivated on Columbia blood agar plates (CBA, Becton Dickinson, Heidelberg, Germany) at 37 °C and 5–10% CO_2_ for 5 days. Yeast-cysteine blood agar plates (HCB, Becton Dickinson, Heidelberg, Germany) were used to cultivate anaerobic bacteria at 37 °C for 10 days (anaerobic chamber, Genbox BioMérieux SA, Marcy/Etoile, France). The number of CFUs per cm^2^ was determined using a colony counter (WTW BZG 40, Xylem Analytics, Weilheim, Germany). All measurements were repeated twice.

### 4.5. DAPI (4′,6-Diamidine-2′-phenylindole Dihydrochloride) Staining

The root-filling materials and the control material were stained after the initial in situ adhesion with DAPI which visualizes all adherent microorganisms under the epifluorescence microscope. A stock solution of DAPI (1 mM) was prepared in a phosphate-buffered NaCl solution (PBS, Biochrom GmbH, Berlin, Germany). The staining solution consisted of 6 µl of the stock solution, which was diluted in 12 mL PBS. The material samples were incubated with 1 mL DAPI (Merck, Darmstadt, Germany) and the staining was conducted in a dark chamber for 10 min. Afterward, the DAPI solution was removed by rinsing with PBS and dried at room temperature. To quantify the total bacterial number per cm^2^, the same procedure was conducted using the inverse epifluorescence microscope (ApoTome.2, Axio Observer.Z1, ZEISS, Oberkochen, Germany) with a 63 × oil immersion objective (Plan-Apochromat 63x/1.4 Oil DIC, ZEISS, Oberkochen, Germany) as described above for the determination of viable and dead bacteria using live/dead staining.

### 4.6. Live/Dead Staining and Fluorescence Microscopy 

Fluorescent propidium iodide (PI) stain was used with the SYTO^®^ 9 (Live/Dead^®^ BacLight™ Bacterial Viability Kit, Life Technologies GmbH, Darmstadt, Germany) to determine the number of viable and dead bacterial cells [53]. Intact cells and those with disrupted membranes can be penetrated by the green fluorescence stain SYTO^®^ 9, whereas the red-fluorescent PI can only penetrate disrupted cell membranes. Hence, viable and active bacterial cells fluoresce green and non-intact cells fluoresce red. The PI and SYTO^®^ 9 were diluted in 0.9% NaCl to achieve a final concentration of 0.1 nmol/mL. The different materials covered with the initial bacterial adhesion were then transferred to multiwell plates and stained with 1 mL SYTO^®^ 9/PI solution in 0.9% NaCl per well, for 15 min at room temperature, in a dark chamber. The stained materials were subsequently placed with the contaminated side on a drop of 0.9% NaCl solution in an 8-chambered cover glass (µ Slide 8 well, ibidi GmbH, Munich, Germany), and analyzed using an inverse epifluorescence microscope (ApoTome.2, Axio Observer.Z1, ZEISS, Oberkochen, Germany) with a 63 × oil immersion objective (Plan-Apochromat 63x/1.4 Oil DIC, ZEISS, Oberkochen, Germany). 

### 4.7. Image Analysis

To quantify the total adhered bacteria visualized after DAPI staining as well as the vital and non-intact bacterial cells (live/dead staining) on each material, 15 different locations of the initial two-hour biofilms were screened for each material sample and wearing cycle. All samples were analyzed using FM with a 63 × oil immersion objective (ApoTome.2, Zeiss, Oberkochen, Germany). The active and dead cells on the images obtained were quantified using the image analysis program ZEN 2 pro (Zeiss, Oberkochen, Germany), and the coverage rates of the live and dead bacteria were subsequently calculated from the data obtained [53]. The results were also visualized by representative microscopic images that were acquired using a 3-Megapixel Microscope camera (Axiocam 503 mono, ZEISS, Oberkochen, Germany). 

### 4.8. Statistical Analysis

For descriptive analysis, mean values and standard deviations were computed. The Dunntest was used to test for differences between the groups, the method of Benjamini-Hochberg was used to correct for multiple testing. All statistical analyses were performed using the statistical software STATA (Version 17, College Station, TX, USA). The level of statistical significance was set to 0.05.

## 5. Conclusions

Substantial quantitative differences in the initial adhesion of oral microbiota were found for the three different materials tested in the present study. Interestingly, MTA seems to allow for a higher initial bacterial adhesion compared to Biodentine and both materials seem to have similar antimicrobial activity.

## Figures and Tables

**Figure 1 antibiotics-12-00004-f001:**
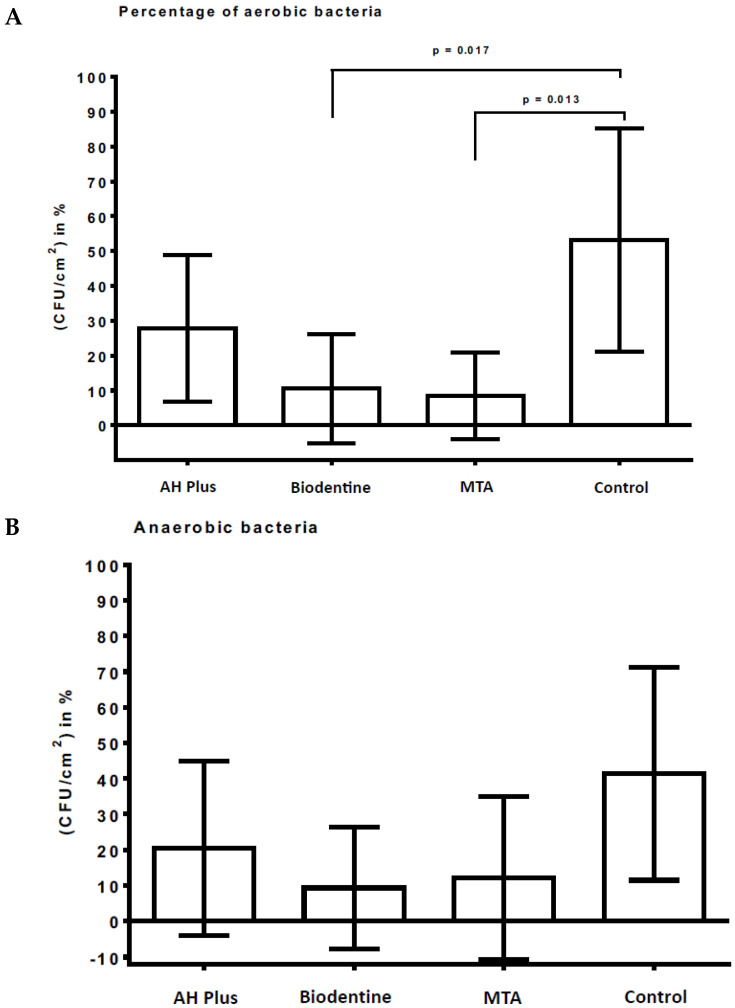
Microbial adhesion and growth of oral microorganisms on AH Plus, Biodentine, MTA, and bovine dentin. The graphs show the number of CFUs which demonstrate the antimicrobial effect of the tested material surfaces on aerobic (**A**) and anaerobic (**B**) bacteria after an exposure time of 2 h in the oral cavity. Bovine dentin served as a negative control, while AH Plus, Biodentine, and MTA served as the material test surfaces. The CFU values were shown on a log_10_ scale per cm^2^ (log_10_/cm^2^). The standard deviations and *p* values of the significantly different data are marked on the graphs.

**Figure 2 antibiotics-12-00004-f002:**
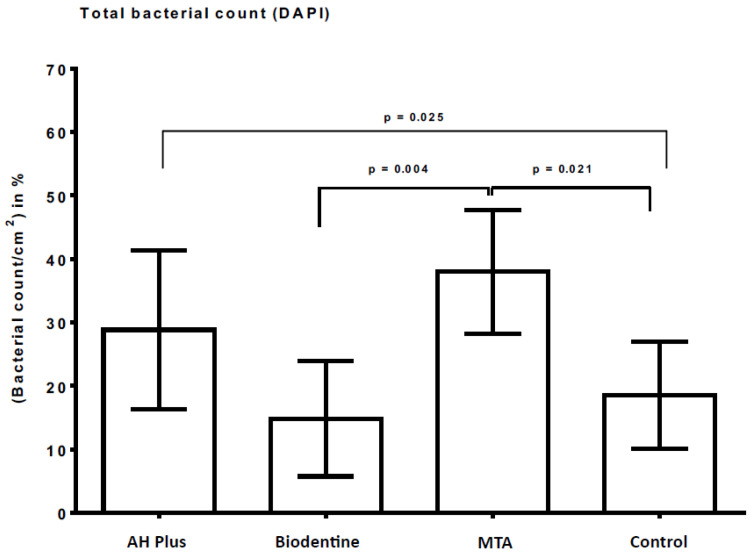
DAPI staining and fluorescence microscopy to visualize adherent microorganisms on AH Plus, Biodentine, MTA, and bovine dentin. The graphs represent the percentage of adherent microorganisms, which were evaluated by the DAPI staining under the fluorescence microscope (FM). Bovine dentin served as a negative control, while AH Plus, Biodentine, and MTA served as material test surfaces. Standard deviations and the *p* values of the significantly different data are marked on the graphs.

**Figure 3 antibiotics-12-00004-f003:**
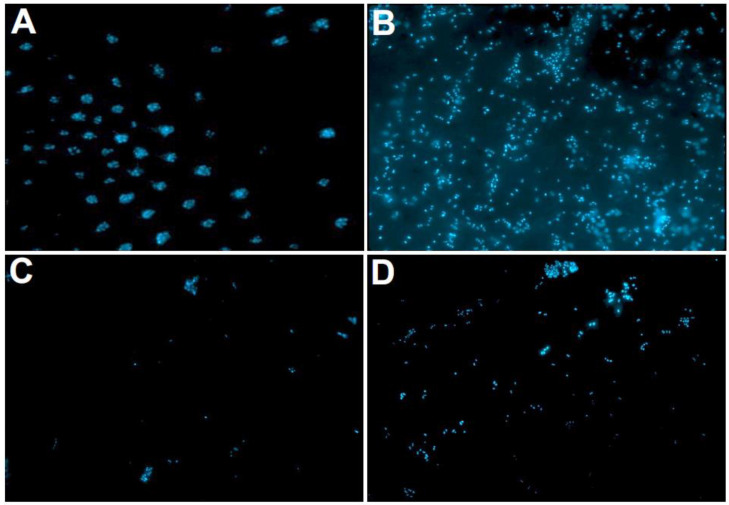
The fluorescence microscopy images after DAPI staining. The initial microbial adhesion (2 h) is depicted on bovine dentin (control) (**A**), MTA (**B**), Biodentine (**C**), and AH Plus (**D**).

**Figure 4 antibiotics-12-00004-f004:**
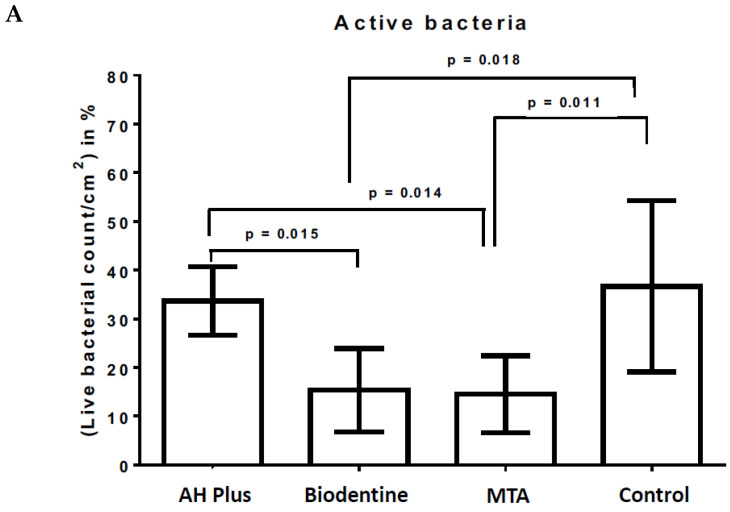
Live/dead staining and fluorescence microscopy. The graphs represent the percentage of “active” oral (**A**) and “non-active” microorganisms (**B**), which were evaluated by the live/dead staining under the fluorescence microscope (FM). Bovine dentin served as a negative control, while AH Plus, Biodentine, and MTA served as material test surfaces. Standard deviations and the *p* values of the significantly different data are marked on the graphs.

**Figure 5 antibiotics-12-00004-f005:**
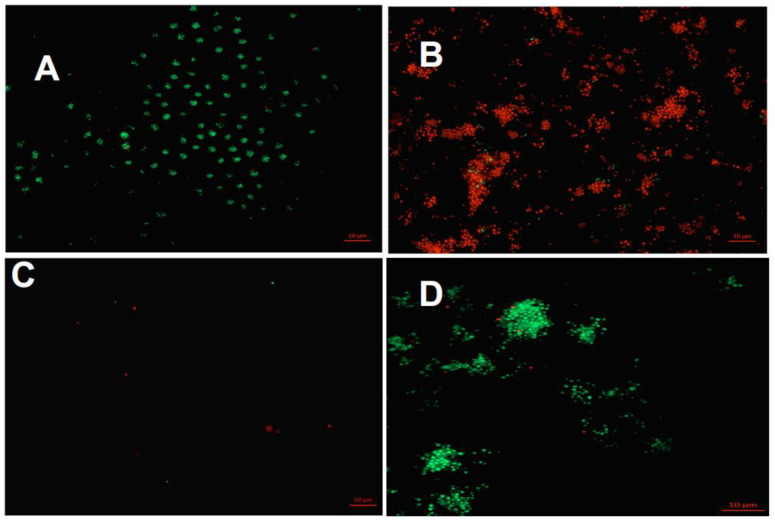
Fluorescence microscopy images after live/dead staining with BacLight^®^. The vital (active) bacteria fluoresce in green, the avital (non-active) in red. The initial adhesion (2 h) is depicted on bovine dentin (control) (**A**), MTA (B), Biodentine (**C**), and AH Plus (**D**).

**Figure 6 antibiotics-12-00004-f006:**
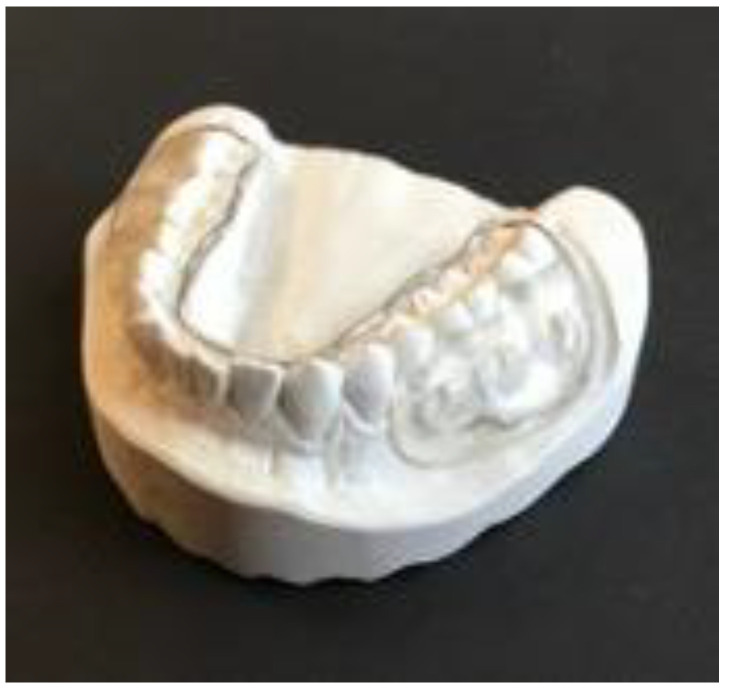
Individually-prepared intraoral lower jaw splint as a carrier for the material samples at the buccal sites of the lower premolars and the first molar.

**Table 1 antibiotics-12-00004-t001:** The distribution of aerobic and anaerobic microorganisms in relation to the slab surface area. The means and standard deviations are depicted, and statistically significant differences are indicated with superscript letters.

	AH Plus	Biodentine	MTA	Control
Anaerobic	27.8% ± 21.2% ^a^	10.5% ± 15.7% ^b^	8.5% ± 12.4% ^b^	53.2% ± 32.0% ^a^
Anaerobic	20.5% ± 24.4% ^a^	9.3 ± 17.1% ^a^	12.2% ± 22.7% ^a^	41.3% ± 29.9% ^a^

**Table 2 antibiotics-12-00004-t002:** The proportion of adherent microorganisms in relation to the surface area of the material and control samples after DAPI staining and fluorescence microscopy.

	AH Plus	Biodentine	MTA	Control
Mean value	29%	15%	38%	18%
SD	13%	9%	10%	9%

**Table 3 antibiotics-12-00004-t003:** The proportions of active and non-active microorganisms in relation to the surface area of the material and control samples after live/dead staining and fluorescence microscopy.

	AH Plus		Biodentine		MTA		Control	
	Active	Non-Active	Active	Non-Active	Active	Non-Active	Active	Non-Active
Mean value	34%	11%	15%	17%	14%	68%	37%	4%
SD	7%	4%	8%	8%	8%	12%	18%	4%

## Data Availability

Data supporting reported results can be found at the corresponding author.

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
