# Peer review of "Biodentine Inhibits the Initial Microbial Adhesion of Oral Microbiota In Vivo"

_antibiotics, 2022, doi:10.3390/antibiotics12010004_

Round 1
Reviewer 1 Report
Comments and Suggestions for Authors
The paper is very well written and contributes a method for evaluating the in vivo initial microbial adhesion of oral micro-organisms on the biomaterial Biodentine compared to MTA and AH Plus. Overall, the study suggests that Biodentine has antimicrobial properties that inhibit microbial adhesion, similar to MTA. The manuscript is of potential interest to the broad readers of antibiotics.
There are some problems, which must be solved before it is considered for publication. If the following problems are well-addressed, this reviewer believes that the essential contribution of this paper is important for bioactive material.
· Major review comments:
1. No conclusion part, it looks very strange, please add a conclusion after the discussion.
2. It is not clear from the information provided whether there are any downsides to using Biodentine and MTA as antimicrobial biomaterials. It would be necessary to conduct further research to evaluate the potential downsides of these materials. Some potential downsides to consider may include potential toxicity, biocompatibility, and long-term effectiveness.
3. Further research is needed to determine the optimal conditions for using these materials, such as the appropriate concentration and application method.
4. six volunteers are not enough, studies can be improved by using larger sample sizes, more rigorous experimental methods, and controlling for potential confounding factors.
· Other minor comments:
5. Figures 1, 2, and 4, The labels with 90 degrees are easier to read.
6. Table 3, no bottom line.
Author Response
Thank you very much for your comments. We tried to adapt the manuscript according to your comments as good as possible. Please find a detailed response to your comments as follows:
Reviewer`s comment:
- No conclusion part, it looks very strange, please add a conclusion after the discussion.
Authors` response:
A separate conclusion headline was inserted and the conclusion was stated as follows:
“In summary, substantial quantitative differences in the initial adhesion of oral microbiota were found for the three different materials tested in the present study. Interestingly, MTA seems to allow for a higher initial bacterial adhesion compared to Biodentine and both materials seem to have similar antimicrobial activity.”
Reviewer`s comment:
- It is not clear from the information provided whether there are any downsides to using Biodentine and MTA as antimicrobial biomaterials. It would be necessary to conduct further research to evaluate the potential downsides of these materials. Some potential downsides to consider may include potential toxicity, biocompatibility, and long-term effectiveness.
Authors` response:
We thank the reviewer for this remark. We included the following paragraph to the discussion:
A limitation using silicate cements is the long setting time of up to 3 h for MTA and approximatly 45 min for Biodentine [8, 35]. The presence of water is necessary for setting and there is a potential of MTA being washed out whenever there is a communication between the oral cavity and the perforation [36]. Using calcium silicates in endodontics and as a dentine replacement, discolorations of the regarding tooth may occur [37]. The mechanism of tooth discoloration was found to be a result of oxidation of the heavy metal oxides (i.e. iron, bismuth or manganese) contained in calcium silicate cements [38]. Although the color change was greater with gray MTA, both, gray and the modified formula of white MTA without iron and magnese induced clinically perceptible crown discoloration [39]. Biodentine contains Zirconium oxide as radiopacifier and doesn`t cause tooth discoloration. However, in contact with sodiumhypochlorite, chlorhexidine or blood discolorations may occur [37]. Additionally, retreatment of endodontic treatments in cases when calcium silicate cements were used as root canal filling material is very difficult [1]. Further research is necessary to evaluate the appropriate concentration, application method, material layer thickness and possibly necessary pretreatments of the application area for the bioactive materials. There is no report on the long-term clinical performance of these tricalcium silicate cement sealers in the literature.
Reviewer`s comment:
- Further research is needed to determine the optimal conditions for using these materials, such as the appropriate concentration and application method.
Authors` response:
Thank you for your comment. We included the following sentence:
“Further research is necessary to evaluate long term effectiveness of the bioactive materials and appropriate concentration, application method, material layer thickness and possibly necessary pretreatments of the application area.”
Reviewer`s comment:
- six volunteers are not enough, studies can be improved by using larger sample sizes, more rigorous experimental methods, and controlling for potential confounding factors.
Authors` response:
We included the following sentence:
“In the present study, a total of six participants carried the intraoral splints as performed previously [47]. However, future studies should aim for a lager sample size since would improve the validity of the findings and potential cofounding factors.”
Reviewer`s comment:
- Figures 1, 2, and 4, The labels with 90 degrees are easier to read.
- Table 3, no bottom line.
Authors` response:
The labels for Figure 1, 2 and 4 were adapted for easier readability.
We inserted a bottom line in Table 3.
Reviewer 2 Report
Comments and Suggestions for Authors
This is an interesting manuscript about the biodentine Inhibits the Initial Microbial Adhesion of Oral Microbiota In Vivo. This manuscript is very well written, organized and I consider of interest in the scientific area.
All comments of this manuscript are for improve the quality of this manuscript for the acceptance.
Introduction section. All information show is enough for this section.
In figure 5C, the image is not clear.
4.1.1. Inclusion criteria. Provide more specific more details, for example: caries free, write teeth caries free or individuals with caries free. age ≥ 18 years, individuals or subjects with age ≥ 18 years. Same for exclusion criteria.
In discussion section, please to discuss the biofilm formation, since the microorganisms in oral cavity are in this form.
In conclusion section, please specific the clinical implication of this study.
Author Response
We thank the reviewer for the important comments. We tried to include all comments/issues in the manuscript and revised it accordingly. Please find a step-by-step-response as follows:
Comment1: Introduction section. All information show is enough for this section.
Response1: Thank you very much for this compliment.
Comment2: In figure 5C, the image is not clear.
Response2: Figure 5 shows representative images of live-dead-stainings. Letters A, B, C and D the images of the different groups indicate the different groups. Image C shows only few red dots and one or two green dots meaning only a small number of adherent microorganisms on the material Biodentine, most of them dead. The description of the active and non-active proportions can be found in ll. 122-137. If the reviewer or editor need additional information according to figure 5, we will provide it.
Comment3: 4.1.1. Inclusion criteria. Provide more specific more details, for example: caries free, write teeth caries free or individuals with caries free. age ≥ 18 years, individuals or subjects with age ≥ 18 years. Same for exclusion criteria.
Response3: Thank you for your comment. We provided more specific information for inclusion and exclusion criteria as follows: “Inclusion criteria: individuals age ≥ 18 years, caries-free individuals; exclusion criteria: individuals with known allergy to the materials or their components and/or suffering from infectious or life-threatening diseases, breastfeeding or pregnant individuals, individuals with temporary use of antibiotics in the last six months or anti-inflammatory medication within the last 30 days, individuals with serious general illnesses, such as diabetes, HIV, hepatitis B and C, acute tumor diseases, or epilepsy.”
The changes are found in the manuscript in correction mode
Comment4: In discussion section, please to discuss the biofilm formation, since the microorganisms in oral cavity are in this form.
Response4: Thank you very much for your important comment. The biofilm formation and influencing factors are very important for our study. We included a paragraph on discussing influencing factors on oral biofilm formation on materials, especially biomaterials. We added the following paragraph: “Initial biofilm formation is affected by multiple factors, e.g. surface charge, surface energy, roughness and topography [29]. Most bacteria have been found to be negatively charged. thus, in general, bacteria are prone to a positively charged surface, while a negatively charged surface is more resistant to bacterial adhesion. Surfaces containing certain cationic groups have antimicrobial activities and thus can kill the attached microbial cells [30]. The setting of silicate cements results in e negatively charged surface charge through an excess of calcium hydroxide formed by OH− leading to a rise in the calcium concentration in the surrounding environment [31]. Roughness and topography seem to be the most important factors regarding microbial adhesion.Thus, smoothening the surface, as we performed in the present study, can reduce biofilm formation and a roughness of Ra of 0.2 µm seems to be the threshold for maximum reduction of bacterial adhesion on surfaces [32]. However, the exact effects of surface roughness on bacterial adhesion and biofilm formation vary with the size and shape of bacterial cells and other environmental factors.”
Comment5: In conclusion section, please specific the clinical implication of this study.
Response5: Thank you for this comment. We added the following sentence to the conclusion indicating the clinical implication of this study: “Biodentine might be the preferred materials used in deep tooth cavities or the pulp chamber, when contaminated with microorganisms.”
Round 2
Reviewer 1 Report
Comments and Suggestions for Authors
Thanks for your revision, I agree to publish.
Author Response
Thank you for your approval.